# Two-Stage Evolution of Gamma-Phase Spherulites of Poly (Vinylidene Fluoride) Induced by Alkylammonium Salt

**DOI:** 10.3390/polym14183901

**Published:** 2022-09-18

**Authors:** Tatsuaki Miyashita, Hiromu Saito

**Affiliations:** 1KF Products Department, Kureha Corporation, Tokyo 103-8552, Japan; 2Department of Organic and Polymer Materials Chemistry, Tokyo University of Agriculture and Technology, Tokyo 184-8588, Japan

**Keywords:** poly(vinylidene fluoride), alkylammonium salt, crystallization, nucleation agent, γ-phase spherulite, light scattering

## Abstract

We investigated the evolution of the γ-phase spherulites of poly(vinylidene fluoride) (PVDF) added to 1 wt% of tetrabutylammonium hydrogen sulfate during the isothermal crystallization at 165 °C through polarized optical microscopy and light scattering measurements. Optically isotropic domains grew, and then optical anisotropy started to increase in the domain to yield spherulite. Double peaks were seen in the time variation of the Vv light scattering intensity caused by the density fluctuation and optical anisotropy, and the Hv light scattering intensity caused by the optical anisotropy started to increase during the second increase in the Vv light scattering intensity. These results suggest the two-stage evolution of the γ-phase spherulites, i.e., the disordered domain grows in the first stage and ordering in the spherulite increases due to the increase in the fraction of the lamellar stacks in the spherulite without a change in the spherulite size in the second stage. Owing to the characteristic crystallization behavior, the birefringence in the γ-phase spherulites of the PVDF/TBAHS was much smaller than that in the α-phase spherulites of the neat PVDF.

## 1. Introduction

Poly(vinylidene fluoride) (PVDF) is broadly used for many applications, such as electrospun nanofibers, films for photovoltaic applications, membranes, and sensors, due to its excellent weatherability, chemical resistance, and ferroelectricity [1,2]. The specific electroactive properties of PVDF strongly depend on its crystal phase. PVDF exhibits at least four crystal polymorphs, i.e., orthorhombic α-, β- and δ-phases, and the monoclinic γ-phase [3,4,5,6]. The α- and δ-phases have the trans-gauche–trans-gauche’ (TGTG’) conformation, β-phase has an all-trans (TTTT) planar zigzag conformation, and γ-phase has a sequence of three trans linked to a gauche (TTTGTTTG’) conformation. The β- and γ-phases are polar phases and are widely studied due to their electric properties, such as their ferroelectric and piezoelectric characteristics [2,4].

The crystalline phase of PVDF is strongly affected by processing methods [3,4,7]. In general, the electrically inactive α-phase is obtained through melt crystallization. On the other hand, the electrically active β-phase is obtained through specific processing, for instance, the hot stretching of the α-phase crystallites below 100 °C [8,9], crystallization in polar solvents [10], miscible mixing with PMMA due to hydrogen bonding interactions between hydrogen atoms of PVDF and carbonyl groups of PMMA [11,12], high pressures above 300 MPa [13], and melt crystallization at ultrahigh cooling rates above 2000 K/s [14]. The electrically active γ-phase is generally obtained through melt crystallization at high temperatures of approximately 170 °C [15] or annealing at high temperature above 158 °C [16]. It has been reported that the polar β- or γ-phase can be effectively obtained by adding inorganic filler [17,18,19]. It is also known that the polar phase can be induced by adding ionic salts, such as ionic liquids [20], KBr [21], cetyltrimethylammonium bromide (CTAB) [22,23,24], ionic fluorinated surfactant [25], and onium salt [26,27].

The effect of ionic salts on the γ-phase nucleation of PVDF is explained by ion–dipole interactions between ionic salts and PVDF chains [28,29,30]. FT-IR study reveals the existence of ion–dipole interactions between tetraalkylammonium salt and PVDF in the molten state, suggesting that ion–dipole interaction promotes the formation of trans sequences [24]. By adding ionic liquids to PVDF, ion–dipole interactions introduce trans sequences, which leads to the formation of the β-phase through rapid cooling, and slow cooling and crystallization at high temperatures relax TTTT conformation of the β-phase to a TTTG one, which leads to the formation of the γ-phase [31]. It is considered that ion–dipole interaction is the driving force for trans sequences in the molten state and crystallizes directly to β-phase crystallites, while the TTTG conformation formed through the relaxation crystallizes to γ-phase crystallites. Since the crystallization rate of the γ-phase is much slower than that of the α-phase [32], it can be expected that further relaxation from TTTG to TGTG occurs and produces α-phase crystallites. However, our previous work revealed that ammonium salts, which have excellent dispersibility, accelerated the crystallization and produced only γ-phase crystallites [33]. Thus, ion–dipole interaction and relaxation are insufficient to understand the evolution of the γ-phase crystallites. It has been widely accepted that the γ-phase crystallites are formed through the ion–dipole interactions between alkylammonium salt and PVDF chains. However, details of the γ-phase nucleation and growth of spherulites have not been clarified.

In this study, we investigated the evolution of the γ-phase spherulites of PVDF during isothermal crystallization using polarized optical microscopic observation and light scattering measurements. Here, the γ-phase crystallites were obtained by adding tetrabutylammonium hydrogen sulfate (TBAHS) as an alkylammonium salt in accordance with the large nucleation agent effect on the γ-phase crystallites suggested in our previous study [33]. The characteristic crystallization behavior of the γ-phase spherulites was also discussed in the results of small-angle X-ray scattering measurements and time-resolved FT-IR measurements.

## 2. Experimental

### 2.1. Preparation of the Specimen

Poly(vinylidene fluoride) (PVDF) was obtained from Kureha Corporation, Tokyo, Japan (grade KF1300, Mw = 350,000). Tetrabutylammonium hydrogen sulfate (TBAHS) was purchased from Koei Chemical Company, Ltd., Tokyo, Japan. The melting temperatures of the PVDF and TBAHS were 171 °C and 174 °C, respectively.

PVDF and TBAHS were melt blended at a weight ratio of 99/1 in a twin blade mixer (R60B, Toyo Seiki Co. Ltd., Tokyo, Japan) connected to a controller and motor (Labo Plastmill 4C150-01, Toyo Seiki Co. Ltd., Tokyo, Japan) at 200 °C and at a rotation speed of 60 rpm for 5 min. For polarized optical microscopic observation, light scattering and small-angle X-ray scattering measurement, a film specimen with a thickness of 500 µm was prepared through hot pressing PVDF/TBAHS at 230 °C, and then quickly cooled to 20 °C. For FT-IR measurements, a thin film specimen with a thickness of 20 μm was prepared through the solvent casting of PVDF and PVDF/TBAHS on a thin aluminum plate using dimethyl acetamide as a solvent.

### 2.2. Light Scattering Measurements

The prepared film specimen was melted at 230 °C for 3 min in a hot stage (Shamal hotplate HHP-411V, AS One Corp., Osaka, Japan), and then rapidly transferred into another hot stage (HS82 hot stage and HS1 hot stage controller, Mettler Toledo Inc., Greifensee, Switzerland) set at the crystallization temperature on the stage of the polarized optical microscope and light scattering equipment. The development of the crystalline morphologies during isothermal crystallization was observed by using a polarized optical microscope (BX53, Olympus Corp., Tokyo, Japan) equipped with a CCD camera (DP74, Olympus Corp., Tokyo, Japan). The structure under the polarized optical microscope was observed using an optical microscope equipped with a sensitive tint plate, with an optical path difference of 530 nm under cross polarizers.

A polarized He–Ne laser with a wavelength of 632.8 nm was applied vertically to the film specimen. The scattered light was passed through the analyzer and then onto a highly sensitive charge-coupled device (CCD) camera with 800 × 600 pixels (pco.1600, Tokyo Instruments Inc., Tokyo, Japan). We employed Hv and Vv geometries in which the optical axis of the analyzer was vertical to that of the polarizer and was horizontal to that of the polarizer, respectively. The input data from the CCD camera were stored in a personal computer for further analysis.

### 2.3. SAXS Measurement

SAXS experiments were performed using the NANO-Viewer system (Rigaku Corp., Tokyo, Japan). Cu-Ka radiation with a wavelength of 0.154 nm was generated at 46 kV and 60 mA, and was collimated using a confocal max-flux mirror system. Measurements were performed at room temperature, and the exposure time was 1 h. An imaging plate (IP) (BAS-SR 127, Fujifilm Corp., Tokyo, Japan) was used as a two-dimensional detector to obtain scattering images. The obtained scattering images were transformed into text data using an IP reading device (RAXIA-Di, Rigaku Corp., Tokyo, Japan). The scattering intensities were corrected with respect to the exposure time, the thickness of the specimen and the transmittance.

### 2.4. FT-IR Measurements

FT-IR measurement was carried out using an infrared spectrometer (FTIR-4100, JASCO Corp., Tokyo, Japan) equipped with an infrared microscope (IRT-5000, JASCO Corp., Tokyo, Japan). To monitor the IR spectra during the crystallization at a high temperature, a hot stage (HS82 hot stage and HS1 hot stage controller, Mettler Toledo Inc., Switzerland) was set on the sample stage of the infrared microscope. A thin film specimen casted on the thin aluminum plate was set on the hot stage, and the time-resolved IR measurement was carried out by averaging 32 scans at a resolution of 8 cm^−1^ with 2 min intervals during the isothermal crystallization at the crystallization temperature of 165 °C after melting at 230 °C.

## 3. Results and Discussion

### 3.1. γ-Phase Spherulite of PVDF/TBAHS

Figure 1 shows the polarized optical micrographs of neat poly(vinylidene fluoride) (PVDF) and PVDF added to 1 wt% tetrabutylammonium hydrogen sulfate (TBAHS) obtained through isothermal crystallization at 165 °C. Large typical spherulites with bright blue and yellow interference colors due to the large birefringence were formed in the neat PVDF. On the other hand, small spherulites with blue-violet and deep pink light interference colors due to the small birefringence were formed in the PVDF/TBAHS. Our previous study revealed that the γ-phase crystallites were obtained and the crystallization of PVDF was accelerated through the addition of alkylammonium salt consisting of short-chain and small-scale anion species, such as TBAHS, due to the nucleation agent effect, while α-phase crystallites were obtained in the neat PVDF [33]. The characteristic band of the α-phase was observed at 1210 cm^−1^ in the neat PVDF, while that of the γ-phase was observed at 1228 cm^−1^ in the PVDF/TBAHS, as shown in the FT-IR spectra (Appendix A). Thus, the small spherulite with small birefringence of the PVDF/TBAHS shown in Figure 1b was γ-phase spherulite, while the large spherulite with large birefringence of the neat PVDF shown in Figure 1a was α-phase spherulite. Since the degree of chain arrangement of the TTTGTTTG’ in the γ-phase is higher than that of the TGTG’ in the α-phase, it is considered that the birefringence of the crystalline chain in the γ-phase is larger than that in the α-phase one. Hence, the small birefringence in the γ-phase spherulite might be attributed to the low ordering in the spherulite due to a low degree of arrangement for lamella in the lamellar stacks or that for lamellar stacks in the spherulite.

Figure 2 shows the small angle X-ray scattering (SAXS) profiles of the α-phase spherulites of the neat PVDF and the γ-phase spherulites of the PVDF/TBAHS obtained through isothermal crystallization at 165 °C. Though the size and birefringence of the α-phase spherulite and γ-phase one were quite different, as shown in Figure 1, the difference in the peak position *q*_m_ and broadness of the peaks was small, indicating that the difference in the periodicity and degree of arrangement for lamellae in the lamellar stacks was also small, i.e., the periodicities of the lamellae in the lamellar stacks *d* in the α-phase spherulite and γ-phase one calculated by d = 2π/*q*_m_ were 9.7 nm and 11.4 nm, respectively. Thus, the low degree of ordering in the γ-phase spherulite suggested by the small birefringence shown in Figure 1 is not attributed to the low degree of arrangement for lamellae in the lamellar stacks, but might be due to the low degree of arrangement for lamellar stacks in the spherulite.

### 3.2. Morphological Evolution

Figure 3 shows the morphological evolution of the α-phase spherulites of the neat PVDF during the isothermal crystallization at 165 °C after a temperature drop from 230 °C. No structure was observed before 10 min (Figure 3a,b). Birefringent embryos with a Maltese cross pattern and a bright blue and yellow interference color due to large birefringence appeared at approximately 13 min (Figure 3c) and grew to larger spherulites (Figure 3d,e). The size of the spherulites increased over time and occupied half the space at 30 min (Figure 3f). Such spherulite growth is typical for the crystallization of polymers [34].

On the other hand, different morphological evolution was seen in the γ-phase spherulites of the PVDF/TBAHS, as shown in Figure 4. Optically isotropic embryos with a red-purple interference color due to a lack of birefringence appeared at approximately 4 min, which was earlier than that of the α-phase spherulites of the neat PVDF, suggesting the nucleation agent effect of TBAHS (Figure 4a,b). No birefringent domain grew to a diameter of 9 μm (Figure 4c), though a clear Maltese cross pattern with large birefringence was seen in the α-phase spherulites when it grew to a diameter of 9 μm (Figure 3c). At approximately 10 min, a light blue-violet and deep pink interference color due to the small birefringence appeared in the domains, which became clearer over time, though the domain size of approximately 11 µm did not increase, suggesting that the ordering in the domains increased to yield spherulite with small birefringence without an increase in the spherulite size (Figure 4d,e). To the best of our knowledge, this is the first report of the characteristic morphological evolution of the γ-phase spherulites of PVDF.

### 3.3. Two-Stage Evolution of the γ-Phase Spherulites

To deeply understand the difference in the morphological evolution shown in Figure 3 and Figure 4, the isothermal crystallization behavior was investigated using light scattering measurement. Figure 5 and Figure 6 show a series of Hv and Vv light scattering images during the isothermal crystallization at 165 °C for the α-phase spherulites of the neat PVDF and the γ-phase spherulites of the PVDF/TBAHS, respectively. Here, Vv light scattering is ascribed to the optical anisotropy and density fluctuation, while the Hv one is ascribed to the optical anisotropy. In the α-phase spherulites, a circular pattern appeared in the Vv mode (Figure 5(a2)) and a four-leaf clover pattern appeared in the Hv mode (Figure 5(b2)), indicating the formation of spherulites with radially arranged lamellar stacks [35]. The circular pattern became larger over time in the Vv mode (Figure 5(a3–a5)), and a four-leaf clover pattern became smaller when the intensity increased in the Hv mode during the growth of the spherulite (Figure 5(b3–b5)).

On the other hand, in the γ-phase spherulites, no scattering pattern was observed in the Hv mode at the early stage (Figure 6(b1,b2)), while a circular pattern was observed in the Vv mode (Figure 6(a1,a2)), indicating that the optically isotropic domain is formed and grows over time at an early stage of the crystallization, as suggested by the polarized optical microscopy shown in Figure 4. The intensity of the circular Vv pattern first increase then decreased (Figure 6(a1–a3)), before increasing over time (Figure 6(a4,a5)). A four-leaf clover pattern appeared in the Hv mode during the second increase in the Vv scattering intensity (Figure 6(b4)), and the Hv scattering intensity increased over time (Figure 6(b4,b5)). The appearance of the four-leaf clover pattern indicated the formation of spherulites with radially arranged lamellar stacks as in the case of the α-phase spherulites. These results suggest the two-stage evolution of the γ-phase spherulites, i.e., the optically isotropic domain is formed and grows over time in the first stage, and the optical anisotropy starts to increase when the ordering in the spherulite increases in the second stage.

The crystallization kinetics can be estimated using the integrated light scattering intensity, i.e., the invariant *Q* defined by [36,37,38]
(1)$$Q=\int_{0}^{\infty }I\left(q\right)q^{2}dq$$
where *q* is the scattering vector; *q* = (4π/*λ*) sin(*θ*/2); *λ* and *θ* are the wavelength and scattering angle, respectively; and *I* (*q*) is the intensity of the scattered light at *q*. The invariant in Hv mode *Q*_HV_ is described by the mean square optical anisotropy <*δ*^2^>
(2)$$Q_{\mathrm{Hv}}\propto \;\langle \delta^{2}\rangle =\phi_{s}{\left(\alpha_{r}-\alpha_{t}\right)}^{2}$$
where *φ*_s_ is the volume fraction of the spherulite, and *α*_r_ and *α*_t_ are the radial and tangential polarizabilities of the spherulite, respectively [37]. (*α*_r_ − *α*_t_) in Equation (2) is ascribed to the intrinsic anisotropy of the lamellar stacks and the orientation function for the optical axis of the lamellar stacks. (*α*_r_ − *α*_t_) is highest when the lamellar stacks are radially arranged from the center without fluctuation, and it becomes smaller when the ordering of the arrangement for the lamellar stacks decreases by the increase in the fluctuations. Hence, the ordering in the spherulite can be provided quantitatively by the light scattering intensity in addition to the volume fraction of the spherulite. On the other hand, the invariant in the Vv mode, *Q*_Vv_, is ascribed to both <*δ*^2^> and the mean-square density fluctuation <*η*^2^>. The <*η*^2^> is given by
(3)$$\langle \eta^{2}\rangle =\phi_{s}\left(1-\phi_{s}\right){\left(\alpha_{c}-\alpha_{a}\right)}^{2}$$
where *α*_c_ is the average polarizability of the crystalline domain, and *α*_a_ is the polarizability of the melt.

Figure 7 shows the time evolutions of the invariant *Q*_Hv_ and *Q*_Vv_ during the isothermal crystallization at 165 °C for the α-phase spherulites of the neat PVDF and the γ-phase spherulites of the PVDF/TBAHS obtained from Figure 5 and Figure 6. A single peak was seen in the *Q*_Vv_ over time in the α-phase spherulites (Figure 7a). The *Q*_Vv_ started to increase at 10 min, attained a maximum at 20 min and then leveled off, while the *Q*_Hv_ started to increase at 20 min and then levelled off. A small decrease in the *Q*_Hv_ was seen before the level off due to the multiple scattering owing to the large spherulite. The *Q*_Vv_ attained a maximum when the *Q*_Hv_ reached half the value of that at the completion of the crystallization, as expected from Equation (3), i.e., the *Q*_Vv_ attains a maximum when the volume fraction of the spherulite *φ*_s_ is 50%. Thus, the change in the *Q*_Hv_ and *Q*_Vv_ is attributed to the spherulite growth. A lag in the onset time of *Q*_Hv_ and *Q*_Vv_ was seen due to the low ordering in the spherulite at an early stage of the crystallization, i.e., the (*α*_r_ − *α*_t_) in Equation (2), is small and it increases over time at an early stage of the crystallization, as suggested in poly(ethylene terephthalate) (PET) [38].

The interesting result here is that double peaks were seen in the *Q*_Vv_ over time in the γ-phase spherulites (Figure 7b). The *Q*_Vv_ started to increase at 3 min, attained a first maximum at 7 min and a second maximum at 15 min, and then leveled off at 30 min. On the other hand, the *Q*_Hv_ started to increase at 7 min and leveled off at 30 min without a maximum. The *Q*_Vv_ attained a first maximum before the *Q*_Hv_ started to increase, indicating that the first increase in the *Q*_Vv_ is attributed to only density fluctuation without optical anisotropy. Since the induction period was seen in the increase in the Vv light scattering intensity, the evolution of the density fluctuation is not attributed to the liquid–liquid phase separation via spinodal decomposition, which occurs spontaneously without an induction period of the Vv light scattering intensity [39]. The *Q*_Vv_ attained a second maximum when the *Q*_Hv_ reached half the value of that at the completion of the crystallization. Hence, the second increase in the *Q*_Vv_ is attributed to the increase in the anisotropy in the spherulite, and the second maximum is attributed to the increase in the volume fraction of the crystalline region in the spherulite, i.e., the *φ*_s_ in Equations (2) and (3) for the second stage of the evolution is the volume fraction of the crystalline region in the spherulite. These results support the two-stage evolution of the γ-phase spherulites, i.e., the first maximum of the *Q*_Vv_ is attributed to the growth of the isotropic domain, and the second one is attributed to the increase in the optically anisotropic crystalline region in the spherulite. The increase in the optically anisotropic crystalline region might be attributed to the increase in the fraction of lamellar stacks in the spherulite, as suggested in the late stage of the crystallization in PET [40].

The *Q*_Hv_ was much smaller than *Q*_Vv_ in the γ-phase spherulites of the PVDF/TBAHS (Figure 7b), compared with those in the α-phase spherulites of the neat PVDF (Figure 7a). The small *Q*_Hv_ in the γ-phase spherulites supports the small birefringence in the spherulite shown in Figure 4. This is attributed to the low degree of ordering in the γ-phase spherulites, i.e., the (*α*_r_ − *α*_t_) in Equation (2), is small due to the disordered arrangement of lamellar stacks.

Figure 8 shows the Hv light scattering profiles at an azimuthal angle of 45° during the isothermal crystallization at 165 °C for the α-phase spherulites of the neat PVDF and the γ-phase spherulites of the PVDF/TBAHS obtained from Figure 5 and Figure 6. As expected from the four-leaf clover patterns shown in Figure 5 and Figure 6, the one-dimensional scattering intensity profiles reached a maximum at the scattering vector *q*_max_, as indicated by the arrows. The *q*_max_ shifted to a lower *q* in accordance with the increase in the height of the peak over time in the α-phase spherulites (Figure 8a). The spherulite radius *R* has an inverse relationship with *q*_max_, as described by *R* = 4.09/*q*_max_ [35]. Hence, the results indicate that the spherulite size increased during the crystallization, i.e., the spherulite radius *R* obtained from the *q*_max_ increased from 3.9 to 6.7 μm at the crystallization time from 12.5 to 22 min. On the other hand, the *q*_max_ did not shift over time, while the height of the peak increased in the γ-phase spherulites, indicating that the spherulite size does not change even when the optical anisotropy increases, i.e., the spherulite radius *R* obtained from *q*_max_ is 4.0 μm, and it did not increase during the increase in the optical anisotropy after the growth of the spherulite to *R* = 4.0 μm (Figure 8b). These results support the two-stage evolution in the γ-phase spherulites, i.e., the optical anisotropy in the spherulites increased without an increase in the spherulite size in the second stage of the crystallization after the growth of the isotropic domain in the first stage.

### 3.4. Mechanism of the Two-Stage Evolution of the γ-Phase Spherulites

Figure 9 shows the time-resolved FT-IR spectra of the γ-phase spherulites of the PVDF/TBAHS during the isothermal crystallization at 165 °C. The characteristic band of the γ-phase was observed at 1228 cm^−1^, while that of the α-phase observed at 1210 cm^−1^ was not seen during the evolution from the first stage (0–10 min) to the second one (10–30 min). The peak area at 1228 cm^−1^ increased continuously over time. These results indicate that the γ-phase crystallites are initiated and increase during the evolution from the first stage to the second one without an α-γ solid–solid phase transition. Thus, the characteristic two-stage evolution observed using polarized optical microscopic observation and light scattering measurements is not attributed to the α-γ solid–solid phase transition. Crystallization without solid–solid phase transition is the same as that reported in the PVDF added to ionic liquid [31], while an α-γ solid–solid phase transition occurred in the PVDF added to cetyltrimethyl ammonium bromide [24]. Since a shoulder peak of the γ-phase was seen at 1228 cm^−1^ in the melt state before the crystallization (0 min), it is considered that the γ-phase is initiated without a solid–solid phase transition from the α-phase due to the strong ion–dipole interactions between TBAHS and PVDF chains to promote the formation of trans sequences from the melt state by preventing relaxation from the TTTGTTTG’ conformation in the γ-phase to the TGTG’ in the α-phase, as suggested by Zhu et al. [31].

Figure 10 shows a schematic illustration of the two-stage evolution of the γ-phase spherulites of the PVDF/TBAHS. Disordered γ-phase domains initiated from the melt state grew to larger ones and collided with each other in the first stage (Figure 10a,b). The fraction of the optically anisotropic crystalline region increased to yield the spherulite consisting of radially arranged lamellar stacks by increasing the fraction of lamellar stacks without a change in the spherulite size in the second stage (Figure 10c,d). The two-stage evolution of the γ-phase spherulites might be attributed to the disordered crystalline domains formed by a nucleation agent effect due to the ion–dipole interaction in the first stage and to the delay of the arrangement of the lamellar stacks due to the slow crystallization of the γ-phase crystallites to yield the second stage.

## 4. Conclusions

The polarized optical microscopic observation and light scattering measurement revealed the two-stage evolution of the γ-phase spherulites of the PVDF by adding 1 wt% of TBAHS during the isothermal crystallization at 165 °C. Optically isotropic disordered domains grew in the first stage, and then optical anisotropy started to increase in the domain to yield the spherulite consisting of radially arranged lamellar stacks by increasing the fraction of the lamellar stacks without a change in the spherulite size in the second stage. Such two-stage evolution is not attributed to the α–γ solid–solid phase transition, but might be attributed to the disordered domains formed by the nucleation agent effect due to the ion–dipole interaction in the first stage and to the delay of the arrangement of the lamellar stacks as a result of the slow crystallization of the γ-phase crystallites to yield the spherulites in the second stage. Owing to the characteristic crystallization behavior, the birefringence in the γ-phase spherulites of the PVDF/TBAHS was much smaller than that in the α-phase spherulites of the neat PVDF, though the difference in the arrangement of the lamellae in the lamellar stack was small.

## Figures and Tables

**Figure 1 polymers-14-03901-f001:**
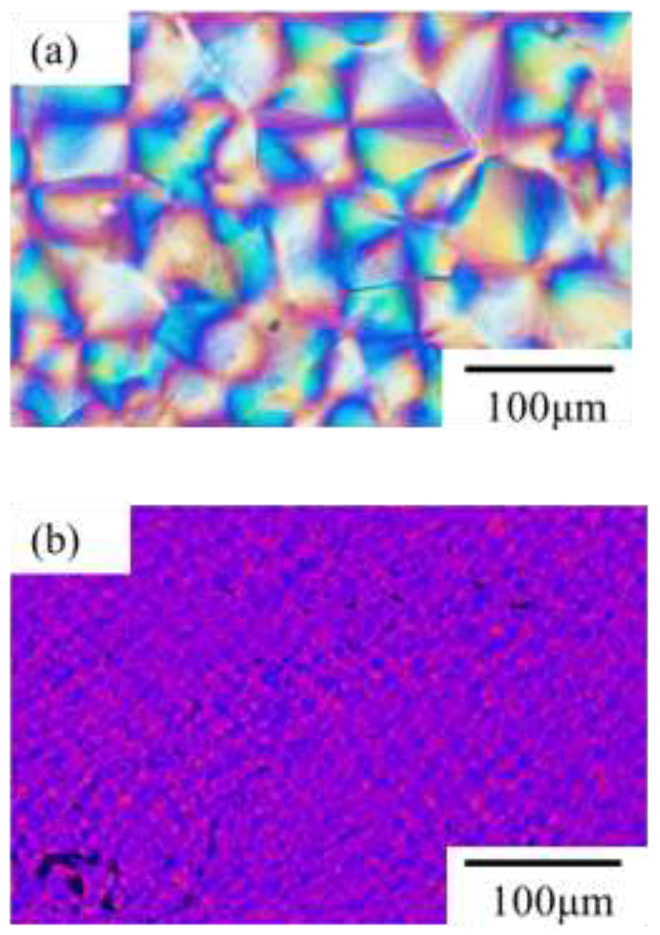
Polarized optical micrographs obtained through isothermal crystallization at 165 °C: (**a**) α-spherulite of neat PVDF; (**b**) γ-spherulite of PVDF/TBAHS.

**Figure 2 polymers-14-03901-f002:**
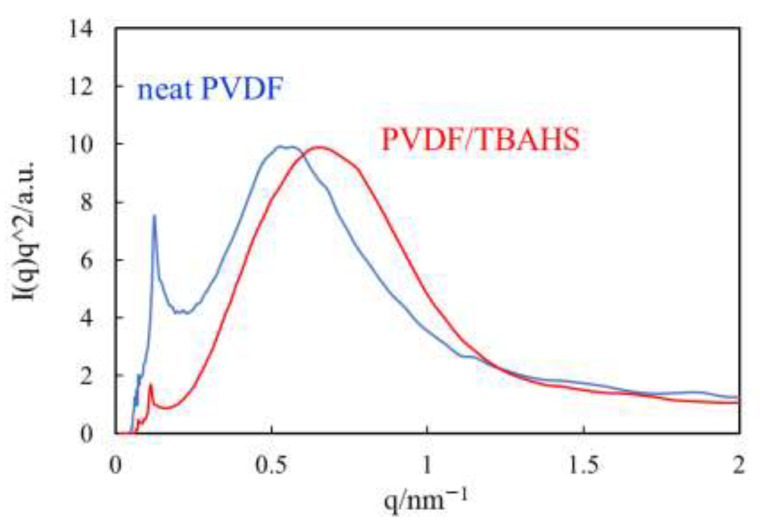
SAXS profiles of α-spherulite of neat PVDF and γ-spherulite of PVDF/TBAHS obtained through isothermal crystallization at 165 °C.

**Figure 3 polymers-14-03901-f003:**
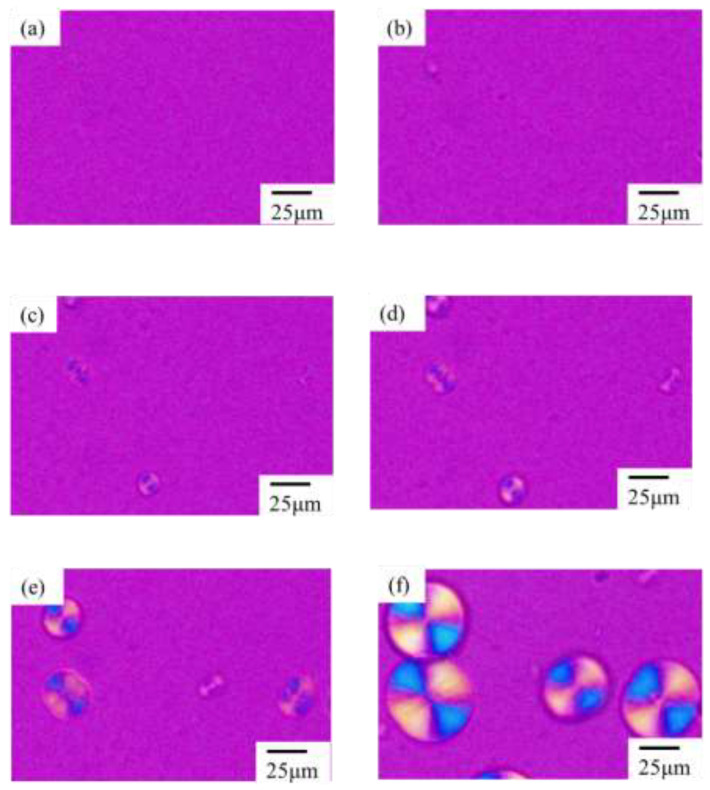
Morphology development of α-spherulite of neat PVDF during isothermal crystallization at 165 °C: (**a**) 8 min, (**b**) 10 min, (**c**) 13 min, (**d**) 15 min, (**e**) 20 min, and (**f**) 30 min.

**Figure 4 polymers-14-03901-f004:**
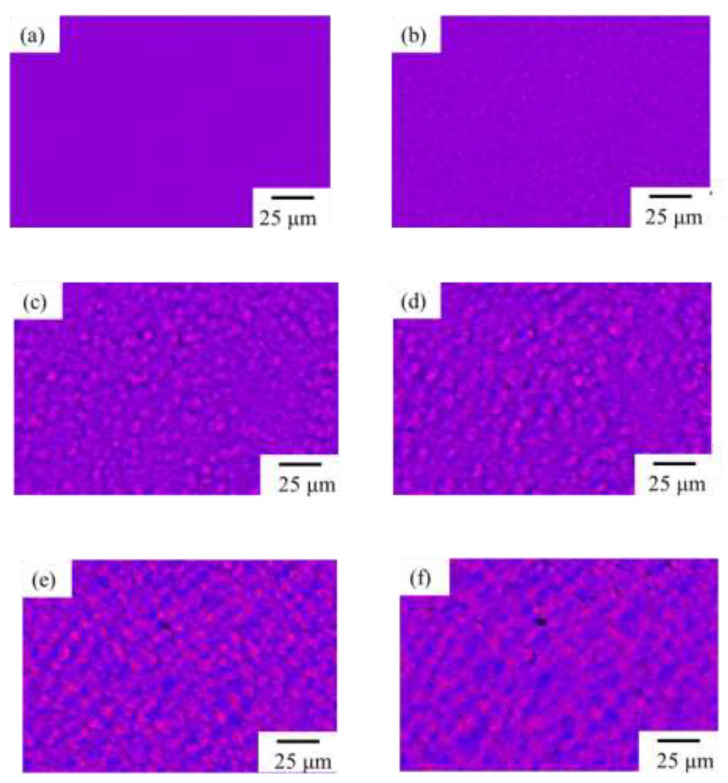
Morphological development of γ-spherulite of PVDF/TBAHS during isothermal crystallization at 165 °C: (**a**) 2 min, (**b**) 4 min, (**c**) 7 min, (**d**) 10 min, (**e**) 15 min, and (**f**) 20 min.

**Figure 5 polymers-14-03901-f005:**
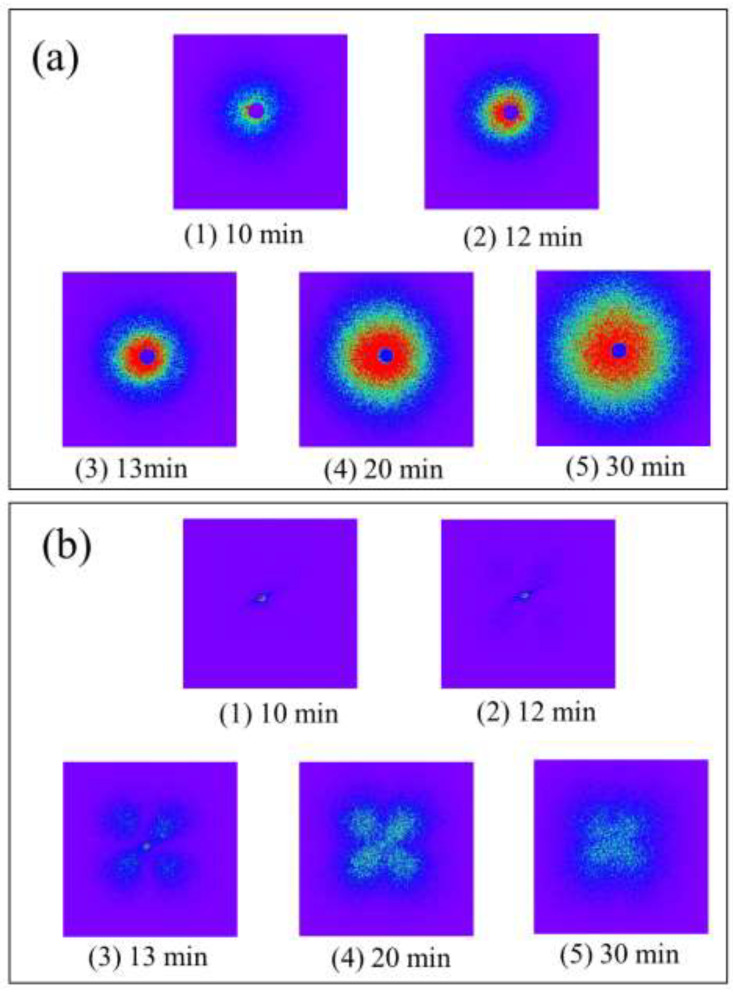
Series of light scattering images of α-spherulite of neat PVDF during isothermal crystallization at 165 °C: (**a**) Vv scattering; (**b**) Hv scattering.

**Figure 6 polymers-14-03901-f006:**
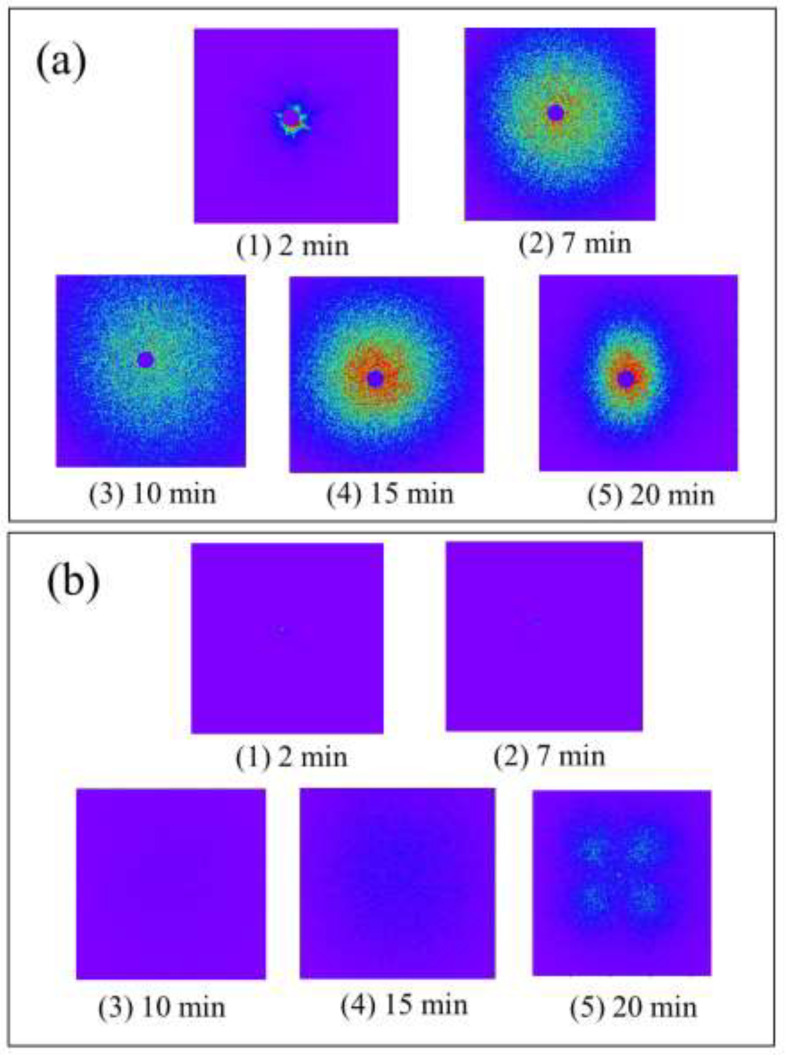
Series of light scattering image of γ-spherulite of PVDF/TBAHS during isothermal crystallization at 165 °C: (**a**) Vv scattering; (**b**) Hv scattering.

**Figure 7 polymers-14-03901-f007:**
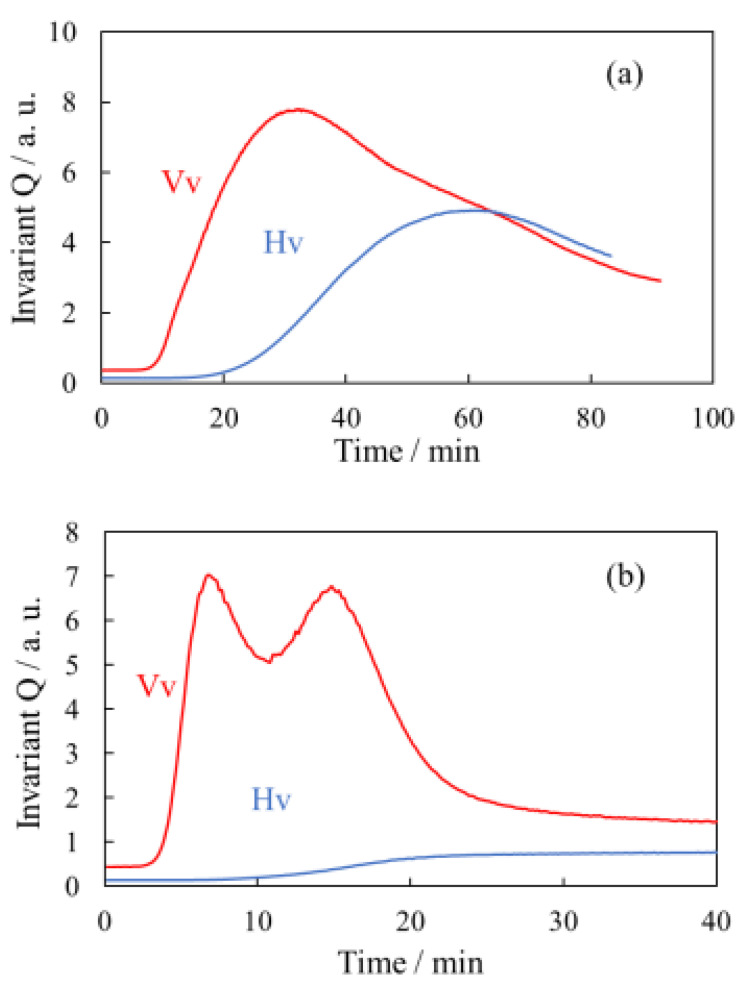
Time evolutions of invariant *Q*_Hv_ and *Q*_Vv_ during isothermal crystallization at 165 °C: (**a**) α-spherulite of neat PVDF; (**b**) γ-spherulite of PVDF/TBAHS.

**Figure 8 polymers-14-03901-f008:**
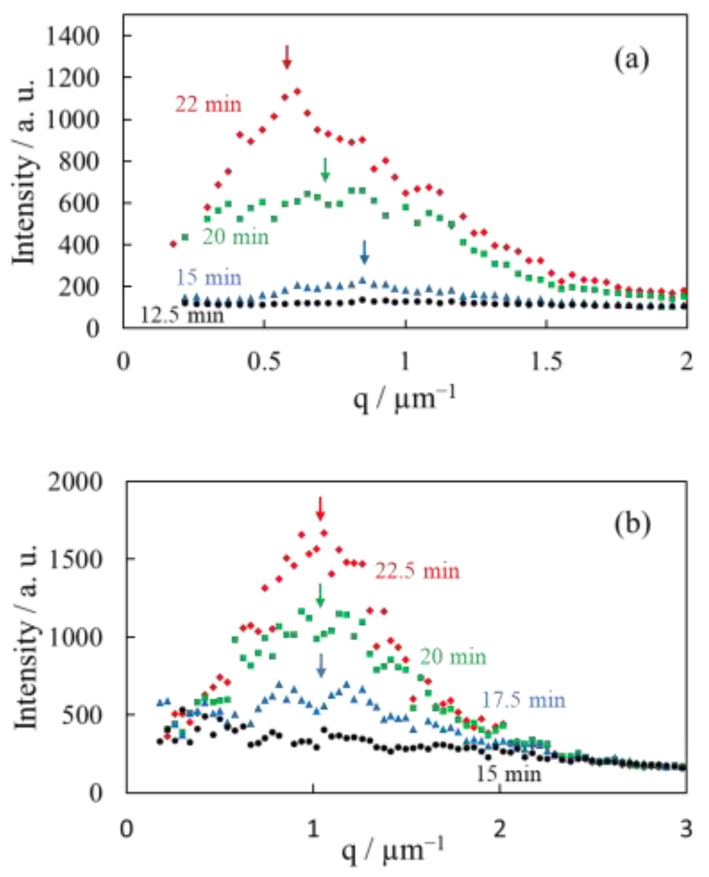
Hv light scattering profiles at an azimuthal angle of 45° during isothermal crystallization at 165 °C: (**a**) α-spherulite of neat PVDF; (**b**) γ-spherulite of PVDF/TBAHS.

**Figure 9 polymers-14-03901-f009:**
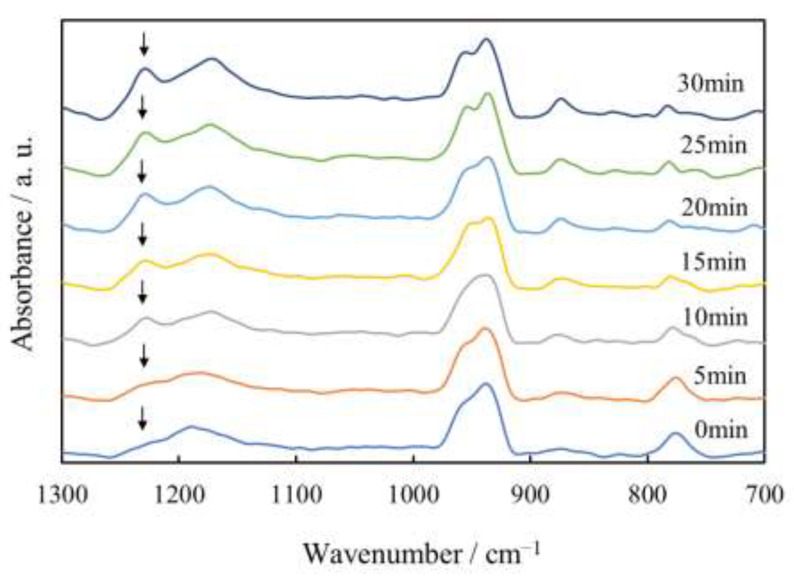
Time resolved FT-IR spectra of γ-spherulite of PVDF/TBAHS during isothermal crystallization at 165 °C.

**Figure 10 polymers-14-03901-f010:**
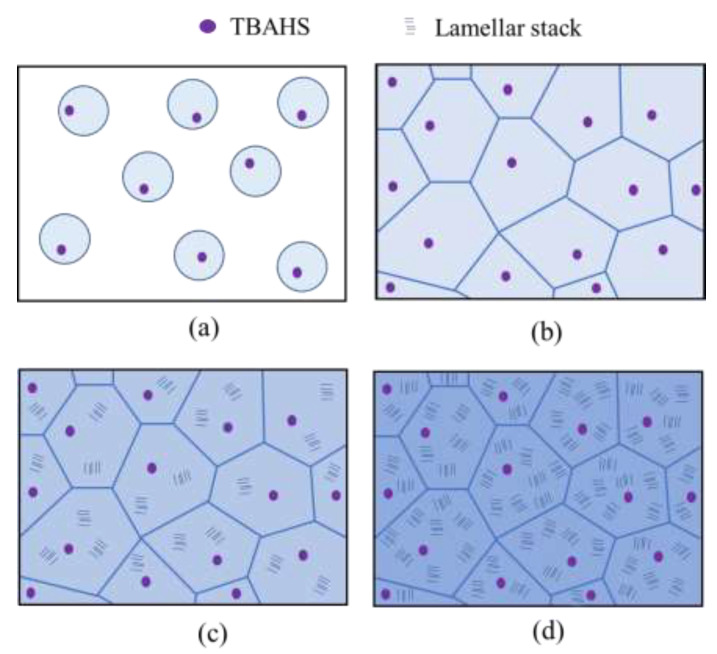
Schematic illustrations of the development of γ-spherulite of PVDF/TBAHS during isothermal crystallization at 165 °C: (**a**) 4 min, (**b**) 10 min, (**c**) 15 min, and (**d**) 20 min.

## Data Availability

Not applicable.

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
