# Peer review of "Two-Stage Evolution of Gamma-Phase Spherulites of Poly (Vinylidene Fluoride) Induced by Alkylammonium Salt"

_polymers, 2022, doi:10.3390/polym14183901_

Round 1

Reviewer 1 Report

Dear authors,

Comments:

1. It is necessary to add clarifying data. Heat elongation - it is probably better to write hot stretching; specify due to what the beta phase is formed during mixing; specify the values of pressure, high temperature, ultra-high cooling rate.

2. If there is information, the stability of the beta and gamma phases should be noted. Is there a reverse transition to the alpha phase.

3. lines 48-52. it would be better to choose either the mention of phases or the corresponding types of conformations.

4. It is not clear what the "low arrangement" or "arrangement is higher" means.

5.Lines 156-157. This method of determining the degree of crystallinity raises huge doubts. The WAXS methodology should be used for this purpose.

6.It is necessary to clearly formulate how the ordering degree was determined. In general, there is confusion in the understanding of order / disorder. The first in the classical sense refers to the crystalline phase, the second - to the amorphous one. The arrangement of crystals can be random or oriented in a specific direction. Maybe it's worth clarifying this and discussing it in the Experiment section

Author Response

Thank you very much for reviewing our manuscript and your valuable comments to improve our manuscript. Following your comments, we revised the manuscript. The poor English in this manuscript was reviewed by MDPI English editing service and extensively corrected. The revised parts based on the reviewer comments are marked in yellow to distinguish them from the English editing which was marked by using the "Track Changes" function. The replies to your comments are listed in the following.

Point 1: It is necessary to add clarifying data. Heat elongation - it is probably better to write hot stretching; specify due to what the beta phase is formed during mixing; specify the values of pressure, high temperature, ultra-high cooling rate.

Response 1: Thank you for your suggestion. To clarify the data, specific values were added in Line 39-45 in the revised manuscript.

Point 2: If there is information, the stability of the beta and gamma phases should be noted. Is there a reverse transition to the alpha phase.

Response 2: Thank you very much for valuable question. To our knowledge, there is no information about the stability of the beta and gamma phases. We would like to investigate the stability of these crystalline phases by controlling the temperature in future.

Point 3: lines 48-52. it would be better to choose either the mention of phases or the corresponding types of conformations.

Response 3: Thank you for your suggestion. To prevent the confusion, we revised “TTTT conformation to TTTG one” to “TTTT conformation of the β-phase to a TTTG one”. (Line 57 in the revised manuscript)

Point 4: It is not clear what the "low arrangement" or "arrangement is higher" means.

Response 4: Thank you for your suggestion. To prevent the confusion, we revised “low arrangement” to “low degree of arrangement”.

Point 5: Lines 156-157. This method of determining the degree of crystallinity raises huge doubts. The WAXS methodology should be used for this purpose.

Response 5: Thank you for your comment. Since the peak area of the SAXS profile is ascribed to volume fraction of the spherulite, volume fraction of lamellar stacks in the spherulites, degree of crystallinity in the lamellar stacks, and difference in the electron density between crystal and amorphous phases, it is considered that the peak area is a measure of the crystallinity. The explanation was added in the revised Lines 171-174.

Point 6: It is necessary to clearly formulate how the ordering degree was determined. In general, there is confusion in the understanding of order / disorder. The first in the classical sense refers to the crystalline phase, the second - to the amorphous one. The arrangement of crystals can be random or oriented in a specific direction. Maybe it's worth clarifying this and discussing it in the Experiment section

Response 6: Thank you for your comment. To clarify the ordering, we added “(αr - αt) is highest when the lamellar stacks are radially arranged from a center without fluctuation, and it becomes smaller when the ordering of the arrangement for the lamellar stacks decreases by the increase of the fluctuations.” in the revised Lines 262-264. We also revised “crystalline region” in the previous Lines 232 and 233 to “lamellar stack” in the revised Lines 261 and 262.

Reviewer 2 Report

The evolution of the γ-phase spherulites of poly(vinylidene fluoride) (PVDF) added with 1wt% of tetrabutylammonium hydrogen sulfate during the isothermal crystallization at 165 °C by polarized optical microscopy and light scattering measurements. The research design is appropriate, the conclusions supported by the results and all the cited references relevant to the research. I think this is an interesting and nice study. So I suggest accept the manuscript after minor revise.

(1) English language and style are fine/minor spell check required.

(2)The "low arrangement" and "arrangement is higher" is not clear.

Author Response

Thank you very much for reviewing our manuscript and your valuable comments to improve our manuscript. Following your comments, we revised the manuscript. The poor English in this manuscript was reviewed by MDPI English editing service and extensively corrected. The revised parts based on the reviewer comments are marked in yellow to distinguish them from the English editing which was marked by using the "Track Changes" function. The replies to your comments are listed in the following.

The evolution of the γ-phase spherulites of poly(vinylidene fluoride) (PVDF) added with 1wt% of tetrabutylammonium hydrogen sulfate during the isothermal crystallization at 165 °C by polarized optical microscopy and light scattering measurements. The research design is appropriate, the conclusions supported by the results and all the cited references relevant to the research. I think this is an interesting and nice study. So I suggest accept the manuscript after minor revise.

Point 1: English language and style are fine/minor spell check required.

Response 1: Thank you for your comment. The poor English in this manuscript was reviewed by MDPI English editing service and extensively corrected.

Point 2: The "low arrangement" and "arrangement is higher" is not clear.

Response 2: Thank you for your suggestion. To prevent the confusion, we revised “low arrangement” to “low degree of arrangement. The explanation for the ordering was added in the revised Lines 262-264.
